# Osteonecrosis in Gaucher disease in the era of multiple therapies: Biomarker set for risk stratification from a tertiary referral center

Mohsen Basiri[1], Mohammad E Ghaffari[2], Jiapeng Ruan[1], Vagishwari Murugesan[3], Nathaniel Kleytman[1], Glenn Belinsky[1], Amir Akhavan[4], Andrew Lischuk[5], Lilu Guo[6], Katherine Klinger[6], Pramod K Mistry[1]*

[1]Department of Internal Medicine, Yale University, New Haven, United States; [2]Department of ENT, Head and Neck Surgery, Guilan University of Medical Sciences, Rasht, Islamic Republic of Iran; [3]Department of Rheumatology, University of Toronto, Toronto, Canada; [4]Department of Computer and Information Science, University of Massachusetts Dartmouth, Dartmout, United States; [5]Department of Radiology and Biomedical Imaging, Yale University, New Haven, United States; [6]Translational Sciences, Sanofi, Framingham, United States

*For correspondence:
pramod.mistry@yale.edu

## Abstract

**Background:** A salutary effect of treatments for Gaucher disease (GD) has been a reduction in the incidence of avascular osteonecrosis (AVN). However, there are reports of AVN in patients receiving enzyme replacement therapy (ERT) , and it is not known whether it is related to individual treatments, *GBA* genotypes, phenotypes, biomarkers of residual disease activity, or anti-drug antibodies. Prompted by development of AVN in several patients receiving ERT, we aimed to delineate the determinants of AVN in patients receiving ERT or eliglustat substrate reduction therapy (SRT) during 20 years in a tertiary referral center.

**Methods:** Longitudinal follow-ups of 155 GD patients between 2001 and 2021 were analyzed for episodes of AVN on therapy, type of therapy, *GBA1* genotype, spleen status, biomarkers, and other disease indicators. We applied mixed-effects logistic model to delineate the independent correlates of AVN while receiving treatment.

**Results:** The patients received cumulative 1382 years of treatment. There were 16 episodes of AVN in 14 patients, with two episodes, each occurring in two patients. Heteroallelic p.Asn409Ser GD1 patients were 10 times (95% CI, 1.5–67.2) more likely than p.Asn409Ser homozygous patients to develop osteonecrosis during treatment. History of AVN prior to treatment initiation was associated with 4.8-fold increased risk of AVN on treatment (95% CI, 1.5–15.2). The risk of AVN among patients receiving velaglucerase ERT was 4.68 times higher compared to patients receiving imiglucerase ERT (95% CI, 1.67–13). No patient receiving eliglustat SRT suffered AVN. There was a significant correlation between GlcSph levels and AVN. Together, these biomarkers reliably predicted risk of AVN during therapy (ROC AUC 0.894, p<0.001).

**Conclusions:** There is a low, but significant risk of AVN in GD in the era of ERT/SRT. We found that increased risk of AVN was related to *GBA* genotype, history of AVN prior to treatment initiation, residual serum GlcSph level, and the type of ERT. No patient receiving SRT developed AVN. These

findings exemplify a new approach to biomarker applications in a rare inborn error of metabolism to evaluate clinical outcomes in comprehensively followed patients and will aid identification of GD patients at higher risk of AVN who will benefit from closer monitoring and treatment optimization.

**Funding:** LSD Training Fellowship from Sanofi to MB.

## Editor's evaluation

This study presents valuable findings on the risk factors of avascular osteonecrosis in patients with Gaucher disease. The evidence supporting the claims of the authors is convincing. The work will interest clinicians who treat patients with inborn errors of metabolism.

## Introduction

In Gaucher disease (GD), biallelic mutations in *GBA1* result in defective lysosomal glucocerebrosidase and the cellular accumulation of glucosylceramide (GlcCer) and its downstream metabolite, glucosylsphingosine (GlcSph) (*Mistry et al., 2015*; *van Dussen et al., 2014*). These lipids accumulate classically in tissue macrophages and trigger a cascade of myeloid cell activation and chronic metabolic inflammation (*Orvisky et al., 2002*; *Rigante et al., 2017*; *Nair et al., 2015*; *Nair et al., 2016*).

Classical disease manifestations of GD include hepatosplenomegaly, cytopenia, and complex skeletal disease involving marrow infiltration, bone pain, osteopenia, fragility fractures, and recurrent avascular osteonecrosis (AVN) (*Hughes et al., 2019*; *Grabowski et al., 2019*; *Yeh et al., 2009*). In the neuronopathic forms of the disease (GD type 2 and type 3), there is additionally childhood onset of neurodegeneration, spinal deformity, pectus carinatum, and pulmonary involvement. Remarkably, severely disabling skeletal manifestations, would a priori, will be expected to be associated with commensurate severe involvement of other disease compartment, but no such relationship can be discerned (*Khan et al., 2012*). In a large study from the International Gaucher Registry (ICGG, ClinicalTrials.gov: NCT00358943), occurrence of AVN did not correlate with severity of hepatosplenomegaly, genotype, thrombocytopenia, or extent of elevation of the disease biomarker, chitotriosidase. Only low hemoglobin and previous splenectomy were correlated with risk of AVN (*Khan et al., 2012*). Therefore, clinicians and patients must remain vigilant of risk of AVN whether or not receiving treatment. Fortunately, there has been a reduction in the incidence of AVN since enzyme replacement therapy (ERT) with macrophage mannose receptor (MMR)-targeted imiglucerase became the standard of care in 1990s (*Mistry et al., 2009b*). The incidence of osteonecrosis in untreated GD patients was reported to be 22.8 per 1000 person years (95% CI, 20.2–25.7) of follow-up. In small single center studies, AVN was reported in as many as half of GD patients (*Desnick et al., 1981*; *Deegan et al., 2011*). The introduction of ERT in the 1990s led to reduction in the incidence of AVN to 13.8 per 1000 years of follow-up. Studies have also shown striking reduction in reports of bone pain and bone crisis in the era of therapeutics (*van Dussen et al., 2014*; *Mistry et al., 2017*). Since 2010, two other MMR-targeted ERTs have become available: velaglucerase and taliglucerase (*Cox, 2013*; *Zimran et al., 2013*; *Gonzalez et al., 2013*). Specific impact of newer ERTs on risk of AVN is not known. A recent introduction to first-line therapies for GD1 includes substrate reduction therapy (SRT), using eliglustat, a potent inhibitor of GlcCer synthase. In the extensive clinical trial program spanning more than 10 years, only few episodes of AVN were reported (*Cox et al., 2023*). In the placebo-controlled ENGAGE trial, there was only one case of AVN, which occurred in a patient receiving placebo (*Mistry et al., 2021*).

Despite the success of ERT in reducing the risk of AVN, there are occasional reports of AVN occurring in patients receiving ERT (*Goker-Alpan, 2011*; *Potnis et al., 2019*; *de Fost et al., 2008*). Therefore, it becomes important to understand whether there are identifiable risk factors for AVN, despite receiving GD-specific therapy. Knowledge of such factors will guide physicians in more comprehensive monitoring of patients beyond hematological and visceral disease as well as potentially advance understanding of underlying mechanisms. Prompted by AVN occurring in two pediatric patients despite ERT, we conducted an analysis of longitudinally followed patients at our tertiary national referral center to identify risk factors for AVN among patients receiving GD-specific therapy. We found patients with compound heterozygous p.Asn409Ser/other genotype, compared to p.Asn409Ser homozygous genotype, were at higher risk of AVN. Other risk factors included history of prior AVN

before treatment initiation, type of ERT, and serum level of GlcSph, a validated biomarker of GD. GlcSph is downstream metabolite produced by deacylation of GlcCer via acid ceramidase. GlcSph is a relevant biomarker to study inflammatory manifestations of GD including AVN, as it is known to trigger immune activation and osteoblast dysfunction (*Nair et al., 2015*; *Dekker et al., 2011*; *Murugesan et al., 2016*; *Saville et al., 2020*).

## Methods

### Patients

We conducted a retrospective study of 187 GD patients longitudinally followed from 2001 to December 2021 at our center. Our observational study is approved by Yale's Human Investigation Committee. A total of 155 patients met the inclusion criteria including confirmed diagnosis of GD by enzymatic and molecular testing, known treatment status and date of initiation/switch (imiglucerase, velaglucerase, and eliglustat), known splenectomy status, previous history of AVN and longitudinal data of MRI volumetrics for liver and spleen, CBC, and biomarkers including GlcSph and chitotriosidase.

Diagnosis of GD was based on diminished levels (<10% compared to controls) of acid β-glucosidase activity in peripheral blood leukocytes and full *GBA1* sequencing using a combination of pacBio sequencing, WES, and Sanger sequencing, as described previously (*Drelichman et al., 2021*). Patients were followed every 1–2 years with standard of care evaluations, including MRI to assess organomegaly and marrow infiltration, and laboratory testing (*Charrow et al., 1998*). Volumetric MRI of liver and spleen was converted to multiples of normal with normal liver volume: 2.5% of body weight and normal spleen volume: 0.2% body weight. Patients developing new bone pain were evaluated earlier. Serum samples were collected at each clinic visit to determine biomarker trends for GlcSph and chitotriosidase, as well as other indicators of disease activity.

### Surrogate disease biomarkers

Serum chitotriosidase activities were determined using a method described previously, with slight modifications (*Schoonhoven et al., 2007*). Briefly, assay buffer was adjusted to pH 4.5, and the final concentration of 4- methylumbelliferyl-beta-D-*N*,*N*′,*N*″-triacetylchitotriose fluorogenic substrate was 10 µM. *CHIT1* genotyping was performed to normalize serum levels as described previously (*Boot et al., 1998*). Serum GlcSph levels were measured via liquid chromatography coupled to tandem mass spectrometry (*Murugesan et al., 2016*). Normal healthy control levels were ≤1 ng/ml.

### AVN episodes

Generally, episodes of AVN were associated with new bone pain or exacerbation of chronic bone pain that prompted new visits, earlier than regular 1–2 yearly follow-ups. Occasionally, we found new AVN on MRI scans and on closer questioning patients reported bone pain that they managed symptomatically.

### Statistical analysis

Frequency and percentage were used to describe qualitative data and mean, and standard deviation were used for quantitative data. Mixed-effects logistic regression was applied to analyze repeated-measure data during treatment. In this study, three mixed-effects models were fitted on the data: a logistic mixed-effects model with the response of AVN incidence and two linear mixed models with the responses of chitotriosidase and GlcSph to different therapies. A random intercept was chosen to account for the presence of different quantitative variables and the uniformity of sizes with different numbers for patients (*Detry and Ma, 2016*). To compare the effect of each treatment on the changes of chitotriosidase and GlcSph, a linear mixed model with two-by-two Bonferroni comparisons was used. The structure of logistics mixed model was as follows:

$$\frac{P(Y_i=1|x_i,z_i)}{P(Y_i=0|x_i,z_i)} = \exp(x_i^T \beta + z_i^T u)$$

where $Y_i$ is a binary variable describing the outcome of case i (0 or 1), β is a fixed parameter vector, $x_i$ is a covariate vector for fixed effects, u is a vector of random variables from probability

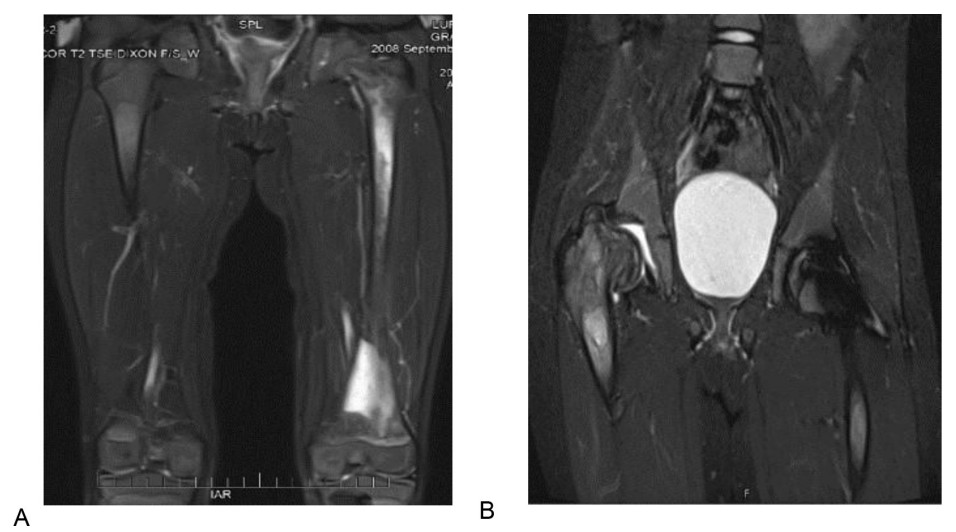

**Figure 1.** T2 weighted and STIR MR images of the patient 1 described in section Illustrative cases: the importance of awareness of AVN in pediatric patients. (**A**) T2 weighted MR image of the femurs, showing diffuse bone marrow signal abnormality throughout the left femur with associated periostitis and subperiosteal fluid collection. (**B**) STIR MR image of the pelvis of the same patient 1 year after. (A) showing an expanding lesion of the greater trochanter, superiorly displacing fragments of bone and collapsed right hip joint.

distributions, and $z_i$ is a covariate vector for random effects. u represents unmeasured covariates as a way of modeling heterogeneity and correlated data.

In the model used in our study, the response variable is incidence of AVN during treatment (yes/no). For fixed effects, the covariates included type of treatment (imiglucerase, velaglucerase, and eliglustat), previous history of AVN (occurrence and non-occurrence), sex, *GBA1* genotype, spleen status (intact spleen with volumetric measurement or splenectomized), and GlcSph, chitotriosidase, hemoglobin, platelet count, spleen volume, liver volume. The random effect was intercept. Several variance-covariance structures were examined for obtaining the best fit. To make this selection, Akaike's information criteria was used for the least value. Autoregressive of Order 1 (Ar (1)) was chosen based on this criterion. All statistical analyses were performed with SPSS version 28 (SPSS Inc, Chicago, IL, USA). A p-value <0.05 was considered statistically significant. MedCalc version 20.026 was used for ROC curve analysis and graphs were plotted by GraphPad Prism version 9.3.1.

## Illustrative cases: the importance of awareness of AVN in pediatric patients

Two illustrative cases underscore the premise for our study. Patient 1 was born in in 2006–2010, had presented at pre-school age with history of chronic epistaxis and easy bruising, and found to have cytopenia and splenomegaly. The proband was diagnosed with GD1, *GBA1* genotype p.Asn409Ser/c.217delC mutations. Liver and spleen volumes were 1.68 and 10.68 multiples of normal respectively. The patient was commenced velaglucerase ERT at dose 60 U/kg/2 weeks. The patient developed infusion-associated reactions (IARs) initially controlled with premedication but became refractory by 24th infusion and was switched to taliglucerase ERT which also caused IARs. Further switch back to velaglucerase caused recurrent IARs. Hematological parameters began to decline and at age 6–10 and developed severe pain in femur concerning for AVN. Plain radiology was normal. Symptoms progressed, and patient was confined to the wheelchair. MRIs revealed diffuse bone marrow signal abnormality throughout the femur consistent with osteonecrosis in left femur (*Figure 1A*). Few months later, the patient developed AVN in right femur (*Figure 1B*). Patient was evaluated at our center 1 year later. The patient appeared chronically ill and in pain, was in wheelchair, and had limited ability to weight-bear. Further studies revealed that patient had developed pan-neutralizing antibodies to all enzyme preparations (velaglucerase, imiglucerase, taliglucerase) after starting velaglucerase ERT. Due to high risk of extension of primary AVN in the femora and recurrent AVN at other sites, we secured

approval from our IRB for treatment with eliglustat via Emergency Use Protocol. During past 6 years on eliglustat, there have been no further bone crises, Hb increased from 10.6 to 13.6, and platelets from 107 to 184. Concomitantly, serum GlcSph was reduced from 644 ng/ml to 101 ng/ml.

### Patient 2

The proband who was in good state of health, started to suffer from left femur pain during school age requiring multiple ER admissions. Initial diagnosis included leg sprain, growing pains, Lyme arthritis, and then osteomyelitis based on MRI findings. Despite antibiotics, femur pain persisted and pathology of debridement for presumed osteomyelitis was positive for Gaucher cells. *GBA1* genotype was p.Asn409Ser/p.Leu483Pro mutations. The proband was started on velaglucerase ERT but few months later, the patient had recurrent AVN in left femur. The patient was switched to imiglucerase and has been remained free of AVN and hematological, visceral, and biomarkers indicators show stable GD activity. More recently after the patient turned young adult, was switched to eliglustat SRT and continued to do well.

## Results

### Patient demographics and AVN on treatment

The demographics of patients in the study are shown in *Table 1*. The cohort comprises 79 (50.3%) male individuals. Of the 155 patients studied, 42 (27.1%) had history of osteonecrosis prior to treatment initiation and 32 patients (20.6%) had undergone prior splenectomy.

During 20 years' span of this study, there were 16 episodes of AVN in 14 patients, with two episodes, each occurring in two patients. In aggregate, the patients received 1382 cumulative years of treatment. By treatment type, total treatment years for imiglucerase ERT was 834 years and a total of six episodes of AVN occurred on imiglucerase ERT, that is 0.72 AVN per 100 years imiglucerase. Patients received cumulative 310 years velaglucerase ERT and there were 10 episodes of AVN, that is 3.2 AVN per 100 years of velaglucerase ERT. Total treatment years for eliglustat SRT was 238 years and so far, there have been no episodes of AVN among patients receiving eliglustat SRT. The demographics of these 14 patients shown in *Table 2*.

### Delineating the determinants of AVN on treatment

Mixed-effects logistic regression was applied for analyzing repeated-measure data during treatment to decipher risk factors for developing AVN on treatment (*Table 3*). Heteroallelic p.Asn409Ser GD patients were 10 times (95% CI, 1.5–67.2, p=0.003) more likely than p.Asn409Ser homozygous patients to develop osteonecrosis during treatment. History of AVN prior to treatment initiation increased the risk of AVN by 4.8-fold while receiving a specific GD treatment (95% CI, 1.5–15.24, p=0.008). The risk of AVN among patients who had received velaglucerase ERT was 4.7 times higher compared to patients receiving imiglucerase ERT (95% CI, 1.67–13.07, p=0.003). No patient receiving eliglustat SRT suffered AVN. There was a significant correlation between residual GlcSph levels and AVN (95% CI, 1.004–1.02, p=0.004). For every 1 ng/ml increase in the level of GlcSph, there was estimated 1.2% increase in the risk of developing osteonecrosis during treatment. Level of serum GlcSph at first and second episode of AVN in two patients further underscores the role of GlcSph for risk prediction. As shown in *Table 2*, patient # 7 had developed first AVN when GlcSph was 85 ng/ml. The second episode of AVN when the serum level of GlcSph was 145.7 ng/ml. Similarly, patient # 8 developed AVN while the serum level of GlcSph was 61.9 ng/ml. The second episode of AVN when the serum level of GlcSph was 77.7 ng/ml.

### Assessing the role of serum GlcSph levels for the risk of AVN

ROC curve analysis was done to estimate cut-offs for GlcSph levels to predict the risk of AVN in GD patients while receiving treatment. The ROC curve shows true positive rate (sensitivity) as a function of the false positive rate (100-specificity) for different cut-off points. We found the value of the area under the ROC curve at GlcSph serum level of 77.64 ng/ml to be 0.857 (*Figure 2A*), that is GlcSph level is a significant variable correlated with the incidence of AVN while receiving treatment.

**Table 1.** Demographic characteristics of the patients in this study.

Among 187 Gaucher disease patients followed longitudinally from 2001 to December 2021, a total of 155 patients met the inclusion criteria (explained in section Patients). Qualitative data were described by frequency and percentage and mean, and standard deviation (SD) were used for quantitative data. *GBA1*: glucosylceramidase beta 1, WBC: white blood cell, HgB: hemoglobin.

| | All (N=155) | | All (N=155) |
|---|---|---|---|
| Age at first visit (year) | | Number of visits | |
| Mean (SD) | 33.81 (18.74) | Mean (SD) | 6.1 (4.17) |
| Median (min, max) | 34 (2, 85) | Median (min, max) | 5.0 (1, 25) |
| Gender | | Duration of follow-up (year) | |
| Female:Male | 77 (49.7 %):78 (50.3%) | Mean (SD) | 14.0 (12) |
| Age at first symptoms (year) | | Median (min, max) | 12(1,20) |
| Mean (SD) | 20.02 (16.1) | Osteonecrosis while untreated | |
| Median (min, max) | 17.0 (5, 65) | Yes | 42 (27.1%) |
| Age at diagnosis (year) | | No | 113 (72.9.%) |
| Mean (SD) | 23.2 (17.3) | Spleen status | |
| Median (min, max) | 22.0 (5, 67) | Intact spleen | 123 (79.4%) |
| Gap to diagnosis in men (year) | | Splenectomized | 32 (20.6%) |
| Mean (SD) | 11.2 (12.6) | Type of treatment (year) | |
| Median (min, max) | 5.5 (0, 57) | Imiglucerase | 834 (60.3%) |
| Gap to diagnosis in women (year) | | Eliglustat | 238 (17.2%) |
| Mean (SD) | 10.3 (11.2) | Velaglucerase | 310 (22.4%) |
| median (min, max) | 7 (0, 40) | Clinical variables: mean (SD) | |
| Age at starting treatment (year) | | Hermann score | 2.84 (1.6) |
| Mean (SD) | 33 (18) | Severity score index | 7.62 (3.8) |
| Median (min, max) | 29 (2, 85) | Chitotriosidase (nmol/hr/ml) | 1106.15 (2801.3) |
| Gap to start treatment (year) | | Glucosylsphingosine (ng/ml) | 58.12 (22.5) |
| Mean (SD) | 10.5 (11) | Liver volume in cc | 1676.40 (186.4) |
| Median (min, max) | 9 (0, 57) | Liver multiples of normal | 0.94 (0.24) |
| *GBA1* genotype | | Spleen volume in cc | 480.9 (146.8) |
| p.Asn409Ser/p. Asn409Ser | 78 (51%) | Spleen multiples of normal | 3.47 (2.95) |
| p.Asn409Ser/p. Leu483Pro | 20 (13.1%) | WBC ×10$^3$ u/l | 6.46 (3.48) |
| p.Asn409Ser/84GG | 14 (9.2%) | Hgb in g/dl | 13.92 (1.6) |
| p.Asn409Ser/IVS2+1 | 6 (3.9%) | Platelets ×10$^3$ u/l | 200.6 (87.1) |
| p.Asn409Ser/other | 27 (17.6%) | | |
| Other/other | 8 (5.2%) | | |

**Table 2.** Demographics and clinical features of patients with osteonecrosis on treatment at the proximity to avascular osteonecrosis (AVN) occurrence.

Modes of presentation of AVN shown as 'painful' indicating new onset of bone pain prompting early evaluation and 'silent' discovered on routine annual MRI and in retrospect, further history indicated patients had exacerbation of chronic pain that they treated symptomatically. * age range: C (child: age <18), YA (young adult: age 18–40), MA (middle-aged adult: age 40–60), OA (older adult: age >60). Shown are GBA1 genotype, age at AVN, type of treatment, history of AVN prior to treatment, serum biomarkers, liver/spleen volumes, and interval between diagnosis and initiation of treatment.

| AVN occurrence | Patient | Gender | Genotype in GBA1 | Age at AVN (category)* | Type of ERT | History of AVN | Age at Diagnosis (category)* | Gap to diagnosis (years) | Age at ERT initiation (category)* | Gap to ERT initiation (year) | Type of AVN | Location | Hemoglobin (gm/dl) | Platelet (×1000 u/l) | Liver multiples of normal | Spleen multiples of normal | Chitotriosidase (nmol/hr/ml) | GlcSph (ng/ml) |
|---|---|---|---|---|---|---|---|---|---|---|---|---|---|---|---|---|---|---|
| 1 | Patient no. 1 | M | p.Asn409Ser /84GG | MA | Velaglucerase | No | C | 1 | YA | 23 | Silent | Femur | 17.1 | 111 | 0.89 | 5.2 | 389.8 | 93.7 |
| 2 | Patient no. 2 | F | p.Asn409Ser /p.Ile442Thr | MA | Velaglucerase | Yes | C | 1 | YA | 11 | Painful | Femur | 15.6 | 252 | 0.93 | 1.8 | 743.5 | 158.3 |
| 3 | Patient no. 3 | M | p.Leu483Pro /p.Asn501Lys | OA | Velaglucerase | Yes | MA | 1 | OA | 25 | Silent | Femur | 14.0 | 313 | 0.80 | Splenectemized | 332.9 | 69.2 |
| 4 | Patient no. 4 | F | p.Asn409Ser /p.Leu483Pro | YA | Imiglucerase | No | C | 0.2 | C | 0.1 | Silent | Femur | 13.3 | 244 | 0.96 | 1.56 | 1962 | 89.8 |
| 5 | Patient no. 5 | F | p.Asn409Ser /p.Leu483Pro | C | Velaglucerase | Yes | C | 3 | C | 0.1 | Painful | Femur | 13.4 | 212 | 1.12 | 3.1 | 731.4 | 143.5 |
| 6 | Patient no. 6 | M | p.Asn409Ser /p.Asp166Glu | MA | Imiglucerase | Yes | MA | 5 | MA | 0.4 | Painful | Shoulder | 15.8 | 90 | 1.82 | 9.1 | 3.4 | 248.2 |
| 7 | Patient no. 7 | F | p.Asn409Ser /p.Arg502Ser | MA | Velaglucerase | No | YA | 10 | YA | 12 | Painful | Femur | 11.6 | 238 | 1.49 | 5.8 | 731 | 85 |
| 8 | Patient no. 7 | | Second episode | | | Yes | | | | | | Painful | Hip | 12.5 | 235 | 1.41 | 5.4 | 733.6 | 145.7 |
| 9 | Patient no. 8 | F | p.Asn409Ser /c.217delC | MA | Imiglucerase | Yes | YA | 0.1 | YA | 9 | Painful | Femur | 11.0 | 222 | 0.8 | Splenectemized | 523.4 | 61.9 |
| 10 | Patient no. 8 | | Second episode | | | Yes | | | | | | Painful | Femur | 10.1 | 322 | 0.75 | Splenectemized | 623.9 | 77.7 |
| 11 | Patient no. 9 | F | p.Asn409Ser /IVS2+1 | OA | Imiglucerase | Yes | C | 7 | YA | 23 | Painful | Hip | 13.3 | 474 | 0.73 | Splenectemized | 563.6 | 128.4 |
| 12 | Patient no. 10 | M | p.Asn409Ser /p.Leu483Pro | MA | Imiglucerase | Yes | YA | 4 | MA | 22 | Painful | Ankle | 16.7 | 299 | 0.88 | Splenectemized | 3123.4 | 223.8 |
| 13 | Patient no. 11 | M | p.Arg463Cys /p.255Tyr | MA | Velaglucerase | Yes | MA | 1 | MA | 5 | Painful | Hip | 14.5 | 253 | 0.91 | 1.6 | 1852.5 | 86.3 |
| 14 | Patient no. 12 | F | p.Asn409Ser /c.217delC | C | Velaglucerase | No | C | 3 | C | 1 | Painful | Femur | 11.5 | 124 | 1.38 | 7.7 | 4134.6 | 318.6 |
| 15 | Patient no. 13 | F | p.Asn409Ser /p.Asn409Ser | MA | Velaglucerase | No | YA | 0 | YA | 10 | Painful | Hand | 13.6 | 189 | 0.83 | 1.5 | 26.2 | 21.9 |
| 16 | Patient no. 14 | M | p.Asn409Ser /84GG | MA | Velaglucerase | Yes | C | 0 | YA | 12 | Painful | Navicular | 13.7 | 147 | 1.12 | 4.15 | 0 | 210.3 |

**Table 3.** Output of mixed-effects modeling.

To determine the clinical context in which avascular osteonecrosis (AVN) occurs in Gaucher disease (GD) patients while receiving enzyme replacement therapy (ERT). Eliglustat substrate reduction therapy (SRT) was not included in the model as there were no episodes of AVN on patients receiving this therapy at the time of analysis. A random intercept was chosen to account for the presence of different quantitative variables and the uniformity of sizes with different numbers for patients. (Further explanation in sections Statistical analysis and Delineating the determinants of AVN on treatment.)

Fixed coefficients

| Model term* | Coefficient | Std. error | t | Sig. | 95% Confidence interval | | Exp(coefficient) | 95% Confidence Interval for Exp(coefficient) | |
|---|---|---|---|---|---|---|---|---|---|
| | | | | | Lower | Upper | | Lower | Upper |
| Intercept | −9.392 | 1.2846 | −7.311 | <0.001 | −11.913 | −6.872 | 8.337E-5 | 6.705E-6 | 0.001 |
| Rx: velaglucerase | 1.543 | 0.5235 | 2.948 | 0.003 | 0.516 | 2.570 | 4.680 | 1.676 | 13.071 |
| Rx: imiglucerase | 0† | . | . | . | . | . | . | . | . |
| AVN pre-Rx: Yes | 1.568 | 0.5894 | 2.659 | 0.008 | 0.411 | 2.724 | 4.795 | 1.508 | 15.243 |
| AVN pre-Rx: No | 0† | . | . | . | . | . | . | . | . |
| Gender: female | 0.859 | 0.5298 | 1.620 | 0.105 | −0.181 | 1.898 | 2.360 | 0.834 | 6.673 |
| Gender: male | 0† | . | . | . | . | . | . | . | . |
| Spleen status: Splenectomized | 0.754 | 0.5851 | 1.288 | 0.198 | −0.394 | 1.902 | 2.125 | 0.674 | 6.697 |
| Spleen status: intact | 0† | . | . | . | . | . | . | . | . |
| Genotype: other/other | 1.903 | 1.3717 | 1.387 | 0.166 | −0.789 | 4.594 | 6.705 | 0.454 | 98.919 |
| Genotype: p.Asn409Ser/other | 2.309 | 0.9685 | 2.384 | 0.017 | 0.409 | 4.209 | 10.063 | 1.505 | 67.297 |
| Genotype: p.Asn409Ser/p.Asn409Ser | 0† | . | . | . | . | . | . | . | . |
| Serum level of glucosylsphingosine (ng/ml) | 0.012 | 0.0040 | 2.880 | 0.004 | 0.004 | 0.020 | 1.012 | 1.004 | 1.020 |

Probability distribution: Binomial Link function: Logit.

*Target: occurrence of AVN during treatment.

†This coefficient is set to zero because it is redundant.

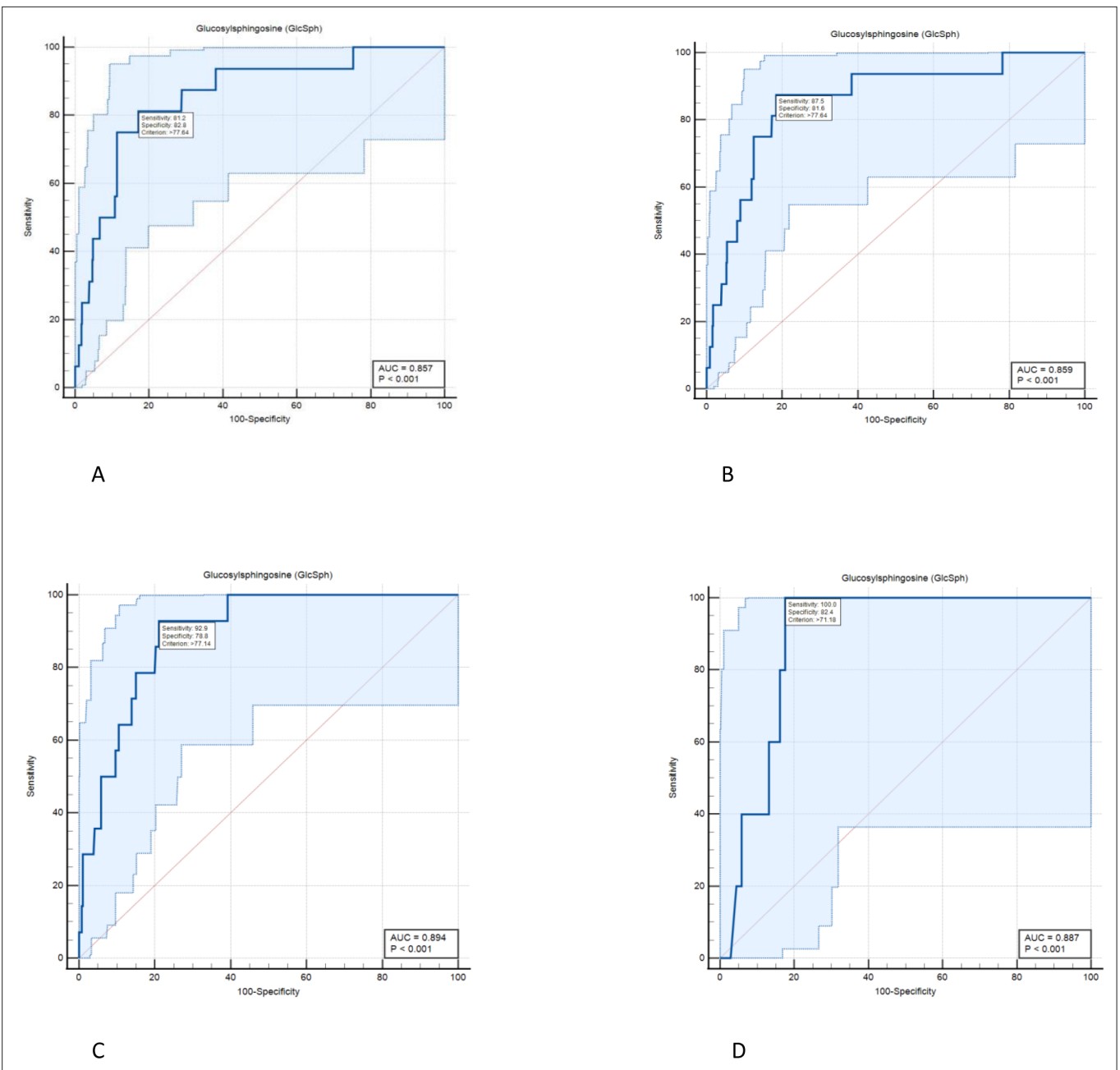

**Figure 2.** Diagnostic accuracy of serum level of glucosylsphingosine (GlcSph) for diagnosis of avascular osteonecrosis (AVN) on treatment: ROC curve analysis was done to assess the diagnostic accuracy cut-offs for GlcSph levels to predict the risk of AVN in Gaucher disease (GD) patients while receiving treatment. (**A**) We found the value of the area under the ROC curve at GlcSph serum level of 77.64 ng/ml to be 0.857. GlcSph level of 77.64 ng/ml has sensitivity of 81.2% (95% CI, 54.4–96%) and specificity of 82.8% (95% CI, 80.7–84.8%) for association with AVN in entire GD patients receiving treatment. (**B**) As shown here, clinical significance of GlcSph levels is enhanced by considering their context of use regarding other risk factors. In patients with at least one of delineated risk factors in our study, that is compound heterozygosity for p.Asn409Ser, history of AVN prior to treatment or velaglucerase ERT, sensitivity for GlcSph level at 77.64 ng/ml increases to 87.5%. (**C**) In patients who harbor at least two of delineated risk factors in this study, GlcSph level of 77.14 ng/ml has sensitivity of 92.9% and specificity of 78.8% to rule in the probability of AVN on ERT. (**D**) In patients with all three risk factors delineated in the study, GlcSph level of 71.8 ng/ml is 100% sensitive to support the probability of AVN with specificity of 82.4%.

## Effect of GD-specific therapies on biomarkers, Chitotriosidase and GlcSph

The three drugs used to treat GD showed differential response in reducing serum chitotriosidase activity. Velaglucerase appeared to be most effective in reducing chitotriosidase, followed by

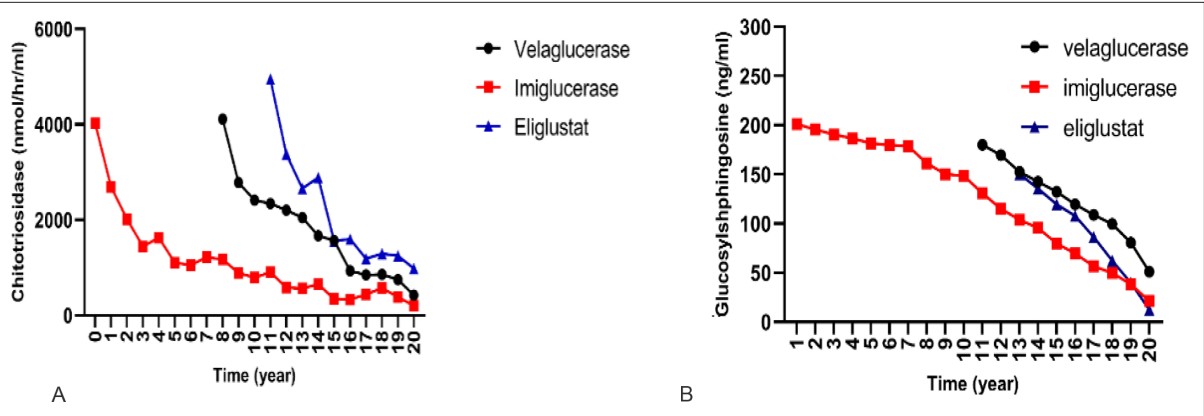

**Figure 3.** Effect of different therapies on chitotriosidase and glucosylsphingosine (GlcSph). To compare the effect of each treatment on the changes of chitotriosidase and GlcSph, a linear mixed model with two-by-two Bonferroni comparisons was used. (**A**) Effects of three drugs on chitotriosidase (nmol/hr/ml): pairwise comparisons showed velaglucerase is most effective in reducing chitotriosidase, followed by imiglucerase, followed by eliglustat. (**B**) Effects of three drugs on GlcSph (ng/ml): three drugs were significantly different for effect on GlcSph reduction (p<0.001). Eliglustat was the most effective in reducing serum GlcSph, followed by imiglucerase, followed by velaglucerase. Pairwise comparisons showed that the mean serum GlcSph in eliglustat-treated patients was significantly lower than the enzyme replacement therapies (ERTs). There was no significant difference in patients receiving velaglucerase vs imiglucerase (p=0.478).

imiglucerase. Pairwise comparisons showed the mean chitotriosidase in patients receiving velaglucerase was significantly lower than eliglustat, however, no significant difference was observed between velaglucerase and imiglucerase (p>0.999), (*Figure 3A*).

Next, we examined serum GlcSph biomarker, and we found the three drugs were significantly different for effect on GlcSph reduction (p<0.001). Eliglustat was the most effective in reducing serum GlcSph, followed by imiglucerase, followed by velaglucerase. Pairwise comparisons showed that the mean serum GlcSph in eliglustat-treated patients was significantly lower than the ERTs. There was no significant difference in patients receiving velaglucerase vs imiglucerase (p=0.478). (*Figure 3B*).

## Discussion

AVN is one of the most devastating and life-altering manifestations of GD that results in chronic disability and need for orthopedic surgeries (*Marcucci et al., 2014*). The underlying mechanism(s) of osteonecrosis in GD is not understood. Several mechanisms have been proposed, including disruption of microcirculation by lipid-laden Gaucher macrophages, abnormal red cell morphology leading to ischemia, and osteocyte death (*Mikosch and Hughes, 2010*). Bone marrow cells in GD exhibit abnormal secretome and osteoblast dysfunction in GD has been linked with accumulating bioactive Gaucher lipid, GlcSph (*Mistry et al., 2010*; *Campeau et al., 2009*). There is also evidence of aberrant osteoclast-osteoblast coupling via reduced sphingosine-1-phosphate in GD (*Ryu et al., 2006*; *Ishii et al., 2009*). Another emerging player in bone cellular pathology of GD is the damage-associated molecular patterns released by necrotic osteocytes via macrophage-inducible C-type lectin (Mincle), which is known to induce osteoclastogenesis and bone loss (*Andreev et al., 2020*). In patients with osteonecrosis, Mincle was highly expressed at skeletal sites of osteocyte death and correlated with strong osteoclastic activity (*Andreev et al., 2020*). GlcCer is a ligand for Mincle, which underscores the complex bone marrow-bone microenvironment in GD and need for optimal therapeutic targeting to prevent disabling AVN.

While AVN occurred frequently in GD in the pre-ERT era (*Mistry et al., 2017*), introduction of ERT since 1991 has reduced its incidence (*Mistry et al., 2009b*; *Mistry et al., 2009a*). However, there are occasional reports of AVN among GD patients receiving ERT (*Goker-Alpan, 2011*; *Potnis et al., 2019*; *de Fost et al., 2008*). Therefore, despite multiplicity of proposed mechanisms underlying AVN in GD, the final pathway likely involves the pathogenic lipids, metabolic inflammation, and lipid-laden Gaucher macrophages. These aberrant pathways are expected to be ameliorated by ERTs and eliglustat SRT but their tissue distribution may have a differential effect (*Mistry et al., 2021*; *Weinreb et al., 2021*), for example small molecule SRT vs recombinant MMR-targeted recombinant ERT.

Our study is the first to critically examine outcomes with respect to types (imiglucerase and velaglucerase) and mode of therapy (eliglustat SRT). In experience garnered over 20 years at a single tertiary referral center involving 1382 cumulative years of GD-specific treatments, we found no episode of AVN among eliglustat SRT-treated patients but there were several episodes among patients receiving imiglucerase and velaglucerase ERT. Unexpectedly, we found compared to imiglucerase ERT, patients receiving velaglucerase ERT had 4.7 odds ratio of AVN (95% CI, 1.67–13.07, p=0.003). The basis for this apparent differential effectiveness to prevent AVN is not known. It should be kept in mind that imiglucerase and velaglucerase are not bioidentical. They differ in glycan residues and have minor amino acid changes. Gene expression analysis in mice infused with these enzymes result in different transcriptional profiles (*Dasgupta et al., 2013*). Moreover, velaglucerase have greater number of mannose residues compared to imiglucerase (*Tekoah, 2013*). Theoretically, greater clearance by the liver and spleen via mannose receptors in macrophages of these organs could result in less delivery to the bone marrow. Additionally, imiglucerase have been shown to reverse osteoblast defect in cell cultures even though osteoblasts do not exhibit mannose receptors (*Panicker et al., 2018*). Our findings underscore the need for investigation of this topic in registry setting such as the Gaucher Outcomes Survey (GOS) that monitor long-term outcomes of velaglucerase ERT (ClinicalTrials.gov: NCT03291223). A phase 4 study on bone outcomes was recently completed, and its results will be relevant (ClinicalTrials.gov: NCT02574286) although the primary end-point for the trial appears to be bone density only. During our study, we did not have any episodes of AVN among patients treated with eliglustat SRT. This is in keeping with strikingly low episodes of AVN in extensive clinical trials (*Cox et al., 2023*). In the ENGAGE placebo-controlled trial, only one episode of AVN occurred in patient on placebo (*Mistry et al., 2021*; *Cox et al., 2023*).

Our results advance optimal management of patients with GD based on their biomarker profile. We found that other significant predictors of AVN while receiving treatment included patients with history of AVN prior to initiation of therapy, p.Ser409Asp/other *GBA* genotype and serum GlcSph level. We did not find any association of splenectomy with occurrence of AVN while on treatment. Some studies have suggested association of AVN with splenectomy (*Mistry et al., 2009b*) but studies in longitudinally followed patients in UK showed AVN occurrence frequently preceded splenectomy suggesting that both AVN and need for splenectomy in pre-ERT era were indicators of severe disease (*Deegan et al., 2011*).

In our cohort, *GBA* genotype had a significant effect on the risk for developing osteonecrosis during treatment. Patients who were heteroallelic for p.Asp409Ser were 10 times more likely than p.Asp409Ser homozygous patients to develop osteonecrosis during treatment. In contrast, p.Asp409Ser/p.Asp409Ser appears to be protective against developing osteonecrosis, underscored by the fact that 51% of our cohort were homozygous for p.Asp409Ser mutation, among whom only one patient developed AVN during treatment. Another significant risk factor for developing AVN on treatment was previous history of AVN prior to treatment initiation.

The biomarkers of GD, chitotriosidase and GlcSph, are reduced by ERT and eliglustat SRT (*Murugesan et al., 2016*). Of the therapies we examined, we found eliglustat to be most effective in reducing

**Table 4.** Serum glucosylsphingosine (GlcSph) levels according to probabilities for avascular osteonecrosis (AVN) occurrence while on treatment with dual cut-offs to stratify patients for AVN risk.

Residual serum GlcSph levels were used to stratify risk of AVN among patients receiving treatment. Our findings permit AVN risk stratification based on GlcSph levels into three groups patients with low risk: GlcSph<21.7 ng/ml, patients with intermediate risk: 21.8ng/ml<GlcSph< 77.64 ng/ml, and patients at high risk: GlcSph>77.64 ng/ml.

| AVN on treatment | GlcSph (ng/ml) | Sensitivity (95% CI) | Specificity (95% CI) | +Likelihood ratio |
|---|---|---|---|---|
| Low risk | <21.7 | 100 (79.4–100) | 24.63 (22.3–27.0) | 1.33 |
| Intermediate risk | 21.8–77.6 | 89 (61.3–98) | 56.9 (54–62.5) | 2.36 |
| High risk | >77.64 | 81.25 (54.4–96) | 82.78 (80.7–84.8) | 4.72 |

serum GlcSph compared to ERTs (imiglucerase>velaglucerase). We did not find any association of residual chitotriosidase and AVN. This is in keeping with other reports (*Khan et al., 2012*). In our study we found residual serum GlcSph to be significantly associated with risk of AVN. Residual GlcSph level refers to serum concentration measured in sample taken at close proximity to the onset of AVN (but not levels prior to initiation of treatment). For the first time, our data demonstrate the significance of residual biomarker level which should aid patient stratification for risk of AVN among patients receiving treatment. We used a dual cut-off strategy for serum level of GlcSph to stratify patients for the risk of AVN while receiving treatment in GD patients. There were no episodes of AVN when GlcSph level was less than 21.7 ng/ml with a sensitivity of 100% (95% CI, 79.4–100%) and specificity 24.6%. In contrast, GlcSph level of 77.64 ng/ml has sensitivity of 81.2% (95% CI, 54.4–96%) and specificity of 82.8% (95% CI, 80.7–84.8%) for association with AVN in GD patients receiving treatment. Therefore, our findings permit AVN risk stratification based on GlcSph levels into three groups (*Table 4*): patients with low risk: GlcSph<21.7 ng/ml, patients with intermediate risk: 21.8 ng/ml<GlcSph<77.64 ng/ml, and patients at high risk: GlcSph>77.64 ng/ml.

These findings suggest that it is prudent to aim to achieve GlcSph levels <21.8 ng/ml as a therapeutic goal, that is, <21.8-fold above normal. In practice, the significance of considering these GlcSph levels is greatly enhanced by considering its context of use regarding other risk factors. For example, as shown in *Figure 2B*, in patients with at least one of delineated risk factors in our study that is compound heterozygosity for p.Asn409Ser, history of AVN prior to treatment or velaglucerase ERT, sensitivity for GlcSph level at 77.64 ng/ml increases to 87.5% (*Figure 2B*). In patients who harbor at least two of these risk factors, GlcSph level of 77.14 ng/ml has sensitivity of 92.9% and specificity of 78.8% to rule in the probability of AVN on ERT (*Figure 2C*). In patients with all three risk factors, GlcSph level of 71.8 ng/ml is 100% sensitive to support the probability of AVN with specificity of 82.4% (*Figure 2D*).

Our findings are not an indication to change ERTs, as we recognize many patients are benefiting from both imiglucerase and velaglucerase. However, our experience suggests that it is unwise for empirical changes in types of ERTs by third parties (*Barranger et al., 2014*). Rather, our findings help to elevate optimal management of GD from prevalent practice of monitoring blood counts with or without biomarkers and occasional imaging to a higher level of risk stratification to identify high-risk patients who would benefit from closer monitoring and treatment optimization. Our study is not without limitations. First, it is a single center study, however, the uniform protocol for comprehensive evaluation and long-term follow-up of individual patients are significant strengths. Second, our center is a tertiary referral center, and a referral bias could have enriched patients with AVN, however, this potential bias would apply across our entire patient population. It will be important to examine the GOS which captures data on outcomes of velaglucerase ERT. Hitherto, reports from GOS have iteratively reported only hematological and visceral outcomes, frequently conflating outcomes from multiple ERTs under a generic term ERT (*Hughes et al., 2022*). It behooves on the investigators to build incremental evidence of efficacy, instead of iterative reports of end-points used in primary trials. It should be noted that in 2014, a study was initiated to assess bone outcomes on velaglucerase ERT ( ClinicalTrials.gov: Identifier: NCT02574286), however, there are no reports to date from this study and end-points do not include reduction of AVN incidence.

In conclusion, our study demonstrates residual risk of AVN on GD patients receiving therapy. Independent risk factors for AVN while patients were receiving therapy were pAsp409Ser/other *GBA* mutation, pre-ERT history of AVN, serum GlcSph level, and velaglucerase ERT. Our results will aid optimal monitoring of GD patients, help enrich clinical trials of rare heterogeneous GD, and stimulate comparative effectiveness analysis in the era of multiple therapies.

## Acknowledgements

We thank our patients for the opportunity to provide their Gaucher-related care and for their participation in our studies. MB is supported by Lysosomal Disease Fellowship from Sanofi.

## Additional information

### Competing interests

Lilu Guo, Katherine Klinger: is an employee of Sanofi and may hold stocks. Pramod K Mistry: Reviewing editor, *eLife*. The other authors declare that no competing interests exist.

### Funding

| Funder | Grant reference number | Author |
|--------|------------------------|--------|
| Sanofi Genzyme | LSD Training Fellowship from Sanofi to MB | Mohsen Basiri |

The funders had no role in study design, data collection and interpretation, or the decision to submit the work for publication.

### Author contributions

Mohsen Basiri, Conceptualization, Data curation, Formal analysis, Validation, Investigation, Visualization, Methodology, Writing - original draft, Writing – review and editing; Mohammad E Ghaffari, Amir Akhavan, Conceptualization, Formal analysis, Validation, Visualization, Methodology, Writing - original draft, Writing – review and editing; Jiapeng Ruan, Resources, Data curation, Methodology, Writing - original draft, Project administration, Writing – review and editing; Vagishwari Murugesan, Data curation, Validation, Methodology, Writing – review and editing; Nathaniel Kleytman, Data curation, Methodology, Project administration, Writing – review and editing; Glenn Belinsky, Data curation, Validation, Methodology, Project administration, Writing – review and editing; Andrew Lischuk, Resources, Data curation, Validation, Visualization, Methodology, Writing – review and editing; Lilu Guo, Katherine Klinger, Resources, Methodology, Writing – review and editing; Pramod K Mistry, Conceptualization, Resources, Data curation, Formal analysis, Supervision, Validation, Investigation, Visualization, Methodology, Writing - original draft, Project administration, Writing – review and editing

### Author ORCIDs

Mohsen Basiri  http://orcid.org/0000-0002-9592-4059
Pramod K Mistry  http://orcid.org/0000-0003-3447-6421

### Ethics

Human subjects: All participants were enrolled in our observational studies approved by Yale's IRB. Patients also were provided with verbal explanations and their data were collected after signing consent forms. HIC#0209021074HIC#1005006783.

### Decision letter and Author response

Decision letter https://doi.org/10.7554/eLife.87537.sa1
Author response https://doi.org/10.7554/eLife.87537.sa2

---

## Additional files

### Supplementary files

- MDAR checklist
- Reporting standard 1. STROBE Checklist.

### Data availability

This observational study is approved by Yale University IRB and each patient provided informed consent. The patients have not provided consent to sharing their data with other investigators. Interested academic, non-commercial researchers can contact the Senior Corresponding author, Dr Pramod Mistry at Pramod.mistry@yale.edu to discuss the request to access original data. They do not need to apply or submit a project proposal. All statistical analyses were performed with SPSS version 28 (SPSS Inc, Chicago, IL, USA). MedCalc version 20.026 was used for ROC curve analysis and graphs were plotted by GraphPad Prism version 9.3.1.We do not have consent to share individual patient data. Even de-identified data risks identification through age and GBA genotype information, thus

violating HIPPA patient confidentiality. Upon request to the Senior Corresponding author, the PI, we will share processed version of datasets.

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
