## [Editor Report]

This study presents valuable findings on the risk factors of avascular osteonecrosis in patients with Gaucher disease. The evidence supporting the claims of the authors is convincing. The work will interest clinicians who treat patients with inborn errors of metabolism.

---

## [Decision Letter]

**Decision letter after peer review:**

Thank you for submitting your article "Osteonecrosis in Gaucher Disease in the era of multiple therapies: biomarker set for risk stratification from a tertiary referral center" for consideration by *eLife*. Your article has been reviewed by 2 peer reviewers, and the evaluation has been overseen by a Reviewing Editor and Mone Zaidi as the Senior Editor.

The reviewers have discussed their reviews with one another, and the Reviewing Editor has drafted this to help you prepare a revised submission. There are only a few comments for you to consider.

*Reviewer #1 (Recommendations for the authors):*

Overall this manuscript is excellent. In addition to the comments in the public review, the following suggestions are noted for improvement:

1) While serum GlcSph levels were stratified according to probabilities for AVN occurrence while on treatment, it would be essential to note the longitudinal follow-up of levels for the two patients with multiple AVN episodes. Is there a δ change in the level that would more dramatically increase a patient's risk relative to an overall cutoff level?

---

## [Author Response]

Reviewer #1 (Recommendations for the authors):Overall this manuscript is excellent. The following suggestions are noted for improvement:

1) While serum GlcSph levels were stratified according to probabilities for AVN occurrence while on treatment, it would be essential to note the longitudinal follow-up of levels for the two patients with multiple AVN episodes. Is there a δ change in the level that would more dramatically increase a patient's risk relative to an overall cutoff level?

During 20 years span of this study, we detected 16 episodes of AVN in 14 patients, with two episodes, each occurring in two patients. Please see Table 2. As shown in Table 2, patient # 7 had developed AVN while the patient was on ERT and the serum level of GlcSph at the time of AVN occurrence was 85 ng/ml. The second episode of AVN occurred a few years later when the serum level of GlcSph was 145.7 ng/ml. Similarly, Patient # 8 developed AVN while the patient was on ERT and the serum level of GlcSph was 61.9 ng/ml. The second episode of AVN occurred a few years later and the serum level of GlcSph was 77.7 ng/ml.

Therefore at least in these two patients there is a significant δ change in serum GlcSph levels at the time of second AVN episode. While it is not possible to specify overall δ change in serum GlcSph levels as a risk factor for AVN, our study clearly show that serum GlcSph is highly significant risk factor for AVN occurrence.